# Effect of Mediterranean Diet in Combination with Isokinetic Exercise Therapy on Body Composition and Cytokine Profile in Patients with Metabolic Syndrome

**DOI:** 10.3390/nu17020256

**Published:** 2025-01-11

**Authors:** Juan A. Suárez-Cuenca, Diana Elisa Díaz-Jiménez, Juan A. Pineda-Juárez, Alondra Gissel Mendoza-Mota, Ofelia Dinora Valencia-Aldana, Said Núñez-Angeles, Eduardo Vera-Gómez, Alejandro Hernández-Patricio, Pavel Loeza-Magaña, Jorge Antonio Lara-Vargas, José Rodolfo Arteaga-Martínez, Ángel Alfonso Garduño-Pérez, Jesús Montoya-Ramírez, María Angélica Díaz-Aranda, Roberto Carlos Chaparro-Hernández, Alberto Melchor-López, Silvia García, José Gutiérrez-Salinas, Paul Mondragón-Terán

**Affiliations:** 1Laboratory of Experimental Metabolism and Clinical Research, CMN “20 de Noviembre”, ISSSTE. San Lorenzo 503, Col. Del Valle, Alcaldía Benito Juárez, Mexico City 03100, Mexico; eduardovera20@gmail.com (E.V.-G.); alejandrospa44@gmail.com (A.H.-P.); yelidiazaranda@gmail.com (M.A.D.-A.); rolasil@yahoo.com.mx (S.G.); 2Hospital General de Zona No. 32 “Dr. Mario Madrazo Navarro”, Instituto Mexicano del Seguro Social, Calzada del Hueso S/N, Col. Santa Úrsula Coapa, Alcaldía Coyoacán, Mexico City 04980, Mexico; 3Hospital General de Zona No. 8 y MF “Dr. Gilberto Flores Izquierdo”, Instituto Mexicano del Seguro Social, Rio Magdalena 289, Col. Tizapan San Ángel, Alcaldía Álvaro Obregón, Mexico City 01090, Mexico; roberto_fcr@hotmail.com (R.C.C.-H.); dralbertomelchor@gmail.com (A.M.-L.); 4Coordinación de Investigación, CMN “20 de Noviembre”, ISSSTE. San Lorenzo 503, Col. Del Valle, Alcaldía Benito Juárez, Mexico City 03100, Mexico; alondramendoza1612@gmail.com (A.G.M.-M.); saidnz@live.com (S.N.-A.); 5Department of Physical Medicine and Rehabilitation, CMN “20 de Noviembre”, ISSSTE. Félix Cuevas 540, Col. Del Valle, Alcaldía Benito Juárez, Mexico City 03100, Mexico; doctor.pavel@hotmail.com (P.L.-M.); ikcaban@yahoo.com.mx (J.R.A.-M.); 6Department of Cardiac Rehabilitation, CMN “20 de Noviembre”, ISSSTE. Félix Cuevas 540, Col. Del Valle, Alcaldía Benito Juárez, Mexico City 03100, Mexico; jramh@yahoo.com.mx; 7Department of Endocrinology, CMN “20 de Noviembre”, ISSSTE. Félix Cuevas 540, Col. Del Valle, Alcaldía Benito Juárez, Mexico City 03100, Mexico; drgarduno@gmail.com; 8Department of Bariatric Surgery, Centro Medico Nacional “20 de Noviembre”, ISSSTE. Félix Cuevas 540, Col. Del Valle, Alcaldía Benito Juárez, Mexico City 03100, Mexico; drjmontoyar@hotmail.com; 9Laboratorio de Bioquímica y Medicina Experimental, CMN “20 de Noviembre”, ISSSTE. Félix Cuevas 540, Col. Del Valle, Alcaldía Benito Juárez, Mexico City 03100, Mexico; quauhtlicutli@yahoo.com; 10Centro de Investigación en Ciencia Aplicada y Tecnología Avanzada Unidad Morelos, Instituto Polítecnico Nacional, Boulevard de la Tecnología, 1036 Z-1, P 2/2, Atlacholoaya 62790, Morelos, Mexico; p.mondragonteran@gmail.com

**Keywords:** Mediterranean diet, isokinetic exercise, body composition, cytokine profile, metabolic syndrome

## Abstract

Background: Metabolic syndrome (MS) is a combination of comorbidities that increase pro-inflammatory cytokines (PIC) production, with subsequent body composition (BC) abnormalities and high cardiovascular risk. Treatment with diet and exercise has been suggested as possible non-pharmacological adjuvant treatment. Objective: To determine changes in BC and PIC in patients with MS after a Mediterranean-type diet (MedDiet) and/or isokinetic exercise (IE). Methods: A controlled randomized clinical trial was conducted at a third-level hospital in Mexico City. Intervention groups: G1: MedDiet + IE; G2: IE; G3: MedDiet; G4: control. Anthropometry, BC, and PIC were collected from the baseline and at 12 weeks post-intervention. MedDiet was estimated from a 24-h recall record. IE consisted of a standard high-intensity anaerobic exercise program. Results: Forty-two patients with MS aged 18–65 years old were included. The most frequent comorbidities were obesity, insulin resistance, and dyslipidemia. After 6 months of intervention, a significant reduction of resistin was observed within the IE group and MedDiet + IE, whereas the former group also showed an increase in adiponectin. Interventions of MedDiet and MedDiet + IE showed a decrease in IL-10. Regarding BC, all groups increased the resistance values in relation to the baseline but were lower than the control group. Conclusions: The results suggest that MedDiet and IE have a selective impact on pro-inflammatory mediators, while the combination of MedDiet and IE may induce only minor changes in BC.

## 1. Introduction

Worldwide, there is a burden in the frequency of obesity and metabolic syndrome (MS). Obesity prevalence has grown up to 6-fold in the last 40 years, and this trend continues to rise. In the USA, the prevalence of obesity is projected to increase to nearly 50% of adults by 2030 [1]. Concomitantly, the prevalence of MS has increased from 37.6% to 41.8% in the USA over the past decade [2], and it ranges between 14% and 50% in other populations, including Latin-American, with higher rates in older age groups and urban populations [3].

Obesity and MS represent comorbidities that include modification of body composition (BC) and an increase in pro-inflammatory cytokines (PIC), which together promote cardiovascular risk [4]. Furthermore, cytokines like IL-1, IL-10, resistin, and adiponectin are involved in adipose tissue metabolism, BC, and regulation of low-degree inflammation, therefore representing potential prognostic/intervention targets between populations with cardiometabolic risk [5,6].

Several interventions have been implemented in order to limit MS and its cardio-metabolic consequences. Non-pharmacological approaches include different type of diets and physical exercise, as well as their combinations. Numerous studies [7,8] support that the Mediterranean diet (MedDiet) is an effective intervention to prevent several chronic diseases and control MS, as well as to modulate PIC. At the same time, “isokinetic exercise” (IE), a particular type of strength training where the speed of muscle contraction is kept constant throughout the entire range of motion, has provided benefits for the control and treatment of MS [9,10].

Although the effects of the MedDiet and IE have been independently described, specific and summary effects of the MedDiet and IE on BC and inflammatory mediators have not been fully described. We hypothesized that each intervention is able to induce different effects on BC and/or PIC, which results in a joined benefit for patients with cardiometabolic risk. Therefore, the present study aimed to determine changes in BC and PIC in patients with MS after Med Diet and/or IE.

## 2. Materials and Methods

A controlled randomized clinical trial was conducted at Centro Médico Nacional “20 de Noviembre”, ISSSTE, Mexico City, Mexico. The study protocol complied with the Declaration of Helsinki and was approved by the Committee of the “Ethics, Research, and Biosafety from Centro Médico Nacional 20 de Noviembre, ISSSTE”, ID approval 083.2018, and was registered at ClinicalTrials.gov, NCT03701425. Signed informed consent was obtained from all participants.

Study population and intervention groups: Patients diagnosed with MS according to NCEP-ATP III classification [11]. Patients were excluded if they had been diagnosed with chronic renal failure (KDIGO 3 or higher), cancer, HIV, chronic heart and/or liver failure and/or COPD, or they had any limitation to perform physical activity or follow a structured exercise plan, at the time of recruitment. Participants were randomized (free software “Research Randomizer” Available online: https://www.randomizer.org (accessed on 11 March 2019)) to receive: (1) Mediterranean diet + isokinetic exercise; (2) isokinetic exercise; (3) Mediterranean diet; (4) control, standard recommended diet, and exercise. The intervention length was 12 weeks. Regarding treatment adherence, good adherence to exercise was considered if the patient attended at least 80% of the scheduled visits to the exercise program; good adherence to the diet was considered if the patient did not vary more than 20% from the caloric recommendation; likewise, diet adherence in the Mediterranean diet group was tested through the questionnaire “MedDiet”.

Data Collection: All patients underwent an initial interview to obtain personal data, MS-related symptoms, time of diagnosis of MS, comorbidities, and pharmacological treatment. Food intake was evaluated through a 24-h diet recall. 

Measurements. All measurements were performed at baseline and after 12 weeks of treatment.

Anthropometry: Anthropometric parameters included weight, height, arm and waist circumferences, and hip circumference. Patients were weighed with a calibrated scale (SECA® model 813 with a maximum capacity of 200 kg and accuracy of ±100 g. SECA Hamburg, Deutschland). Patients were asked to remove all the extra weight they could bring with them to avoid bias at the time of weighing them, they were placed in the middle of the scale, straight standing, with the view to the front and with the arms at the sides as well as the feet approximately at the height of the shoulders. Height was measured with a wall stadimeter (SECA® model 220 with a capacity of 230 cm and an accuracy of ± 1 mm. SECA Hamburg, Deutschland). The patients were placed under the stadimeter straight with their eyes to the front but this time with the arms at the sides and the legs together according to Frankfort’s plan. Likewise, the average circumference of the arm, waist, and hip was measured with SECA metric tape (in cm) (SECA® model 201, SECA Hamburg, Deutschland). For the measurement of muscle strength, a Takei® hand dynamometer (TAKEI Scientific Instruments Co., Ltd., Niigata City, Japan) was used with a measurement range of 0 to 100 kg, 3 consecutive measurements were obtained with the dominant hand and an average of the three measurements was calculated.

Body composition: The body composition was measured by the multi-frequency bioelectrical impedance method with a Body Stat Quad Scan 4000 device (Bodystat® British Isles, England, UK). Briefly, the test was performed in the following conditions: 1-h fasting and no vigorous physical activity or alcohol consumption in the last 24 h previous to measurements. During bioelectrical impedance, patients were asked to avoid metal materials that interfere with the electrical frequencies. Then supine position was adopted, and 4 electrodes were placed, two in the hand and two in the foot (both on the right side). Data obtained were charted according to the BIVA, considering values of resistance and reactance, whereas its location within standardized ellipses and specific quadrants described body composition.

Interventions: Isokinetic exercise was performed following a high-intensity anaerobic exercise program consisting of a 5-min warm-up with active mobilizations, pedaling in a cycle ergometer without resistance or walking in an endless band or in a closed space followed by 5 series of isokinetic exercise with the “November 20” protocol, a 90-s break between sets. In the end, a cooling equal to the warm-up or with stretching was carried out; the program was carried out 3 times a week for a period of 12 weeks. On the other hand, Mediterranean diet was based on pre-established dishes with MedDiet characteristics, as well as equivalent foods, which could be exchanged. The diet plan was individualized according to the patient’s energy expenditure adjusted for age, sex, ideal weight, and height with a distribution of macronutrients of 50% carbohydrates, 30% lipids, and 20% protein divided in five meal times (breakfast, morning afternoon meal, lunch, afternoon meal, and dinner).

Plasma markers and cytokines: Blood samples were obtained by puncture in the peripheral vein, at the beginning and the end of the intervention. After sampling, the samples were placed on ice and centrifuged for 30 min. The plasma obtained was refrigerated at −80 °C until its final processing. The following biochemical variables were evaluated: lipid profile (total cholesterol, LDL cholesterol, HDL cholesterol, triglycerides) and glucose were acquired from the clinical records. Circulating levels of cytokines: IL-1, IL-10, adiponectin, and sarcolipin, were determined by colorimetric ELISA Assay Kits (IL1b cat.BMS224-2TEN; IL-10 cat.BMS215-2TEN, Adiponectin cat.BMS2032-2 ELISA kits Thermo Fisher, Waltham, MA, USA; and sarcolipin cat.027956 Sarcolipin (SLN) BioAssay ELISA Kit US Biological Life Sciences, Waltham, MA, USA); following provider’s instructions. Briefly, a plasma sample is added to the pre-coating microplate with a capture antibody. Then, a secondary antibody, conjugated with an enzyme, is added. Upon adding a substrate, the enzyme produces a color change, which was quantified using a spectrophotometer to determine the concentration of the target molecule.

Statistical analysis: For continuous variables, the results were presented in mean ± standard deviation, or P50 (P25, P75), depending on the normality of distribution according to the Shapiro–Wilk test. Percentage (%) of change of variance and 95% confidence interval (_95%_CI) were also provided). Likewise, categorical variables were resumed as *n* (%). Inferential analyses to evaluate the baseline data were performed either by ANOVA or Kruskal–Wallis tests according to the normality of distribution. The Chi-square test was used to compare categorical variables. After the intervention, the Wilcoxon test and Kruskal–Wallis (with the U-Mann–Whitney U test using the Bonferroni correction if applied) were performed to compare intra- and inter-group differences, respectively. Hotelling’s T2 test was used to evaluate the changes in BIVA vectors (resistance and reactance) in each group. The effect size for continuous variables was estimated by the Cohen D test using the software G*Power® [12]. A *p*-value of <0.05 was considered statistically significant. 

## 3. Results

### 3.1. Clinical–Demographic Characteristics

A total number of 42 patients, aged 55 years old, 78% females, whose most frequent comorbidities were obesity, insulin resistance, and dyslipidemia, constituted the study population. There were non-significant differences between the baseline characteristics between intervention groups (Table 1).

### 3.2. Cytokines

After 6 months of each intervention, there were no modifications of anthropometric variables (Table 2). However, a significant reduction of resistin was observed within the exercise and diet/exercise combination groups, whereas the former group also showed an increase in adiponectin. In addition, the diet and diet/exercise combination groups showed a significant decrease in IL-10, while an increasing trend was observed in the control group (Table 3).

### 3.3. Body Composition

Finally, resistance/reactance curves showed that all groups increased resistance values in relation to the baseline but most significantly in the control group. In addition, resistance/reactance curves showed that all groups increased resistance values in relation to the baselines but most significantly in the control group (Figure 1).

## 4. Discussion

The main finding of the present study was that exercise provided an anti-inflammatory additive effect, mainly given by resistin and adiponectin, whereas MedDiet was related to the reduction of IL-10.

The first analyses showed that our study population was homogeneous, and there was no significant difference between intervention groups. Likewise, the population characteristics and primary outcomes are comparable to other studies [13,14,15,16].

Regarding the effect of interventions on BC, the resistance, a measure denoting the opposition to current given by the amount of cells and water, increased in all groups that performed exercise, suggesting an effect on muscle mass. Interestingly, a lower increase in resistance was observed in the group with the MedDiet, whereas for reactance, other measures of BC given by the opposition to current caused by energy accumulation in cell membranes were significantly increased in the control group. Possible explanations include a higher compliance with the exercise protocol. Consistently, Tinsley et al. demonstrated changes in resistance and reactance values in healthy women who performed resistance training [17], while enhancement of muscle mass has been associated with a reduced risk of all-cause mortality, cardiovascular diseases, and metabolic disorders” [18]. In addition, an increase in body weight observed in the control group may be explained by the non-specific recommendations about diet and exercise given to this group.

Likewise, interventions induced changes in metabolic health, which are further related to cytokine profiles. For instance, a decrease in triglycerides and systolic blood pressure was associated with a concomitant increase in adiponectin and reduction of resistin, respectively, whereas a lack of decrease in glucose was related to a significant reduction of IL-10. Although potential bias related to the small number of patients, interventions induced significant changes in cytokine profiles like resistin, adiponectin, and IL-10 as estimated by medium and large effect sizes, regardless of the sample size. Previous research has reported inconsistent findings on how exercise influences the plasma levels of these adipokines [15,19,20,21], which may be explained by the differences in the type of exercise, body fat distribution, baseline metabolic profile, intervention design, and time point of measurement. 

Exercise-induced reduction of resistin has been consistently documented in several studies, including experimental models and metabolic risk populations. It occurs after moderate to vigorous physical activity, resistance training, or aerobic training [22,23,24]. On the other hand, exercise-induced effect on adiponectin has not been so consistent. Such effect, “if any”, seems to be related to a specific type of exercise characterized by resistance training in combination with aerobic training, which is particularly true for patients with MS [25,26]. 

Furthermore, exercise-induced effects on resistin and adiponectin suggest anti-inflammatory properties and amelioration of insulin sensitivity, respectively. At the same time, changes in IL-10 indicate a concomitant regulation of the immune response [27,28,29]. Such cytokines response denotes a complex interplay that could be considered as protective against MS, diabetes mellitus risk, and cardiovascular diseases, with potential benefits for long-term cardiometabolic health; highlighting advantages for individuals at specific risk. 

Our findings are consistent with other reports stressing the impact of exercise intervention on cytokines and cardiometabolic health. For example, a sedentary routine has been described to induce resistin and leptin as well as reduced plasma levels of adiponectin, which may promote cardiovascular disease, while regular exercise can mitigate such adipokine risk response [27,30]. Likewise, several meta-analyses have evidenced the ability of exercise training to induce a moderate, time&dose-response increase in adiponectin, as well as the reduction of leptin, IL-8, and favorable body mass changes, thereby promoting anti-inflammatory effects, insulin sensitivity, and enhancement of cardio-metabolic health, where individuals with obesity often show more significant changes [19,31,32,33,34]. Additional exercise-driven immunoregulation through IL-10 levels is still complex, and its potential effects deserve further characterization.

On the other hand, MedDiet resulted in the reduction of IL-10, suggesting a decreased anti-inflammatory effect. In this regard, scanty studies have explored the effects of MedDiet on anti-inflammatory cytokines, with heterogeneous results. In vitro stimulation with polyphenols of nutritional relevance induces a reduction of IL-10 during the LPS-triggered release of monocytes from adult subjects, which may support our finding [35]. However, the consumption of a diet supplemented with fish oil ω-3 fatty acids for 4 weeks did not show any modification of plasma IL-10 in overweight, healthy subjects [36], whereas adherence to the MedDiet during FFQ (Food frequency questionnaire) analysis was associated with an increase in anti-inflammatory cytokines like plasma adiponectin [37].

We did not find a significant effect of the MedDiet on PIC. However, other studies have shown that higher adherence to the Mediterranean, again during FFQ analysis, was associated with a decrease in plasma leptin, TNF-α, IL-6, PAI-1, and CRP. Such discrepancies regarding the effect of MedDiet on pro- or anti-inflammatory cytokines may be explained by differences in (1) the study design; (2) the method used to evaluate the influence of the MedDiet; (3) the characteristics of the study population; as well as 4) the quantity and quality of diet consumption [37,38]. 

Some limitations of the present study include the sample size, which may introduce potential bias; additionally, the study site is a reference medical center and may not accurately represent the general cardiometabolic risk population. On the other hand, we consider that exercise-induced changes obtained in the present study are feasible to achieve in a real-world scenario since the OMS-recommended exercise routine, which was performed by the control group, rendered similar results as the isokinetic exercise protocol, in terms of metabolic health, BC and cytokines profile. However, the MedDiet was a bit more difficult to achieve, mainly due to local cultural feeding habits, whereas a tropicalized MedDiet may be a reasonable option. 

## 5. Conclusions

In conclusion, exercise combined with a Mediterranean diet induced changes in cytokines such as resistin, adiponectin, and IL-10, indicating a synergistic anti-inflammatory effect that may contribute to their benefits in reducing cardiometabolic risk.

## Figures and Tables

**Figure 1 nutrients-17-00256-f001:**
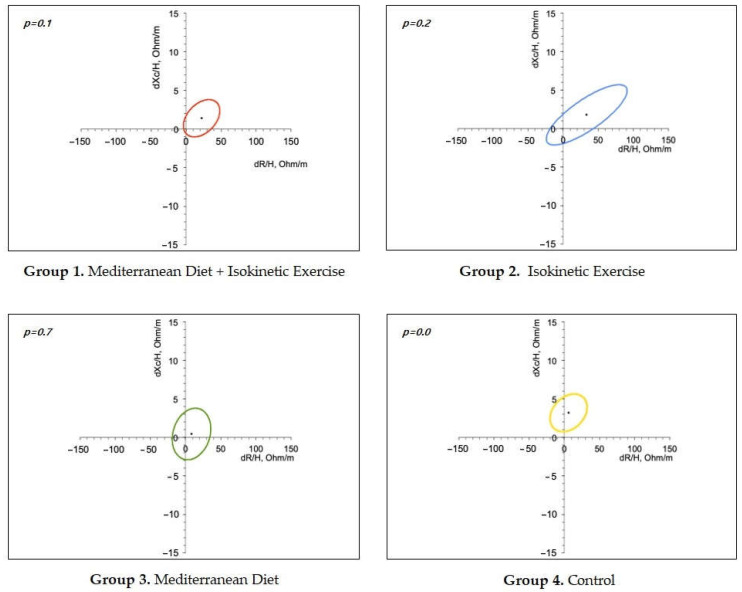
Body Composition. Ellipses graphs show the reactivity and reactance of each treatment group. *p*-values indicate a statistical difference compared to the basal profile.

**Table 1 nutrients-17-00256-t001:** Basal characteristics of the study population (*n* = 42).

	D + E (*n* = 12)	E (*n* = 8)	D (*n* = 14)	C (*n* = 8)	*p*-Value
Age (years-old)	54 (45–56.7)	57 (48–60.7)	56.7 (51–65)	62 (54–67)	0.34
Male Gender *n* (%)	2 (20)	3 (50)	3 (21.4)	1 (12.5)	0.48
Comorbidities *n*(%)					
Insulin Resistance	9 (90)	5 (83.3)	10 (71.4)	7 (87.5)	0.75
Hypertriglyceridemia	6 (60)	5 (83.3)	10 (71.4)	4 (50)	0.61
Hypercholesterolemia	6 (60)	4 (66.7)	11(78.6)	3 (37.5)	0.29
High Blood Pressure	5 (50)	2 (33.3)	11 (78.6)	3 (37.5)	0.14
Obesity	9 (90)	3 (50)	12 (85.7)	8 (100)	0.09
Clinical, Anthropometry					
SBP (mm/Hg)	120 (110–122.5)	110 (107.5–142.5)	130 (117.5–130)	120 (90–130)	0.35
DBP (mm/Hg)	80 (70–82.5)	70 (67.5–82.5)	80 (77.5–92)	80 (80–90)	0.26
Heart Rate (bpm)	66 (64.7–82.2)	62 (57–69)	90 (80–90)	80 (70–90)	0.81
Weight (kg)	90.1 ± 20.6	76.4 ±20.6	83.1 ± 16.2	72.7 +11.3	0.13
BMI (kg/m^2^)	35.4 (28.5–40.3)	28.2 (27.9–35)	31.5 (27.9–37.4)	29.8 (26.0–34.2)	0.45
WC (cm)	107.6 ± 15.3	99.8 ± 5.4	104.6 ± 13.7	97.7 ± 7.1	0.35
HC (cm)	117.9 ± 18.1	106.8 ± 14.5	109.3 ± 12.2	107.7 ± 8.5	0.31
Muscular strength	24.6 ± 6.4	25.6 ± 8	24.6 ± 7.3	19.8 ± 6.6	0.53
Pharmacologic therapy					
Insulin	4 (40)	2 (33.3)	5 (37.5)	2 (25)	0.93
Hypoglycemic	7 (70)	5 (83.3)	9 (63.4)	4 (50)	0.66
Statins	6 (60)	2 (33.3)	8 (57.1)	2 (25)	0.40
Fibrates	2 (20)	1 (16.7)	5 (35.7)	0	0.27
Diuretics	2 (20)	0	5 (35.7)	0	0.13
Antihypertensive	5 (59)	3 (50)	9 (64.3)	2 (25)	0.39
Aspirin	2 (20)	0	4 (28.6)	0	0.25
Biochemical Profile					
Fasting Glucose	128 (121.5–151.5)	100 (93–153)	110 (93.5–147.5)	125.5 (104–147)	0.81
Triglycerides	141.5 (107–176)	115 (101–242)	120 (104–130)	134 (106–166)	0.84
IL-1b	215 (186–249)	200.2 (173–237)	223 (192–313)	280 (220–297)	0.30
Resistin	854 (540–1200)	1457 (966–1630)	1020 (697–1248)	933 (583–1220)	0.21
IL-10	439 (150–481)	419 (300–481)	403 (140–475)	283 (248–505)	0.77
Adiponectin	2.6 (2.1–3.5)	4.2 (2.1–4.5)	2.4 (2.1–5.1)	2.4 (1.7–3.4)	0.79
Sarcolipin	0.38 (0.29–0.61)	0.51 (0.30–0.72)	0.38 (0.33–0.51)	0.27 (0.25–0.58)	0.54

Quantitative data are shown as mean ± DE, or median (p25–p75), and qualitative data are shown as *n* (%). Groups: E, exercise isokinetic; D, Mediterranean diet; D + E, Mediterranean diet + exercise isokinetic; and C, control.

**Table 2 nutrients-17-00256-t002:** Changes in anthropometric characteristics of the study population.

	D + E (*n* = 12)	E (*n* = 8)	D (*n* = 14)	C (*n* = 8)
SBP (mm/Hg)				
Basal	120.0 (110.0, 130.0)	125.0 (110.0, 147.5)	130.0 (112.5, 130.0)	115.0 (92.5, 127.5)
Follow up	120.0 (110.0, 120.0)	110.0 (102.5, 132.5)	120.0 (100.0, 130.0)	115.0 (102.5, 135.0)
% of change	−2.4	0.78	0.65	−9.1
DBP (mm/Hg)				
Basal	80.0 (70.0, 80.0)	70.0 (67.5, 82.5)	80.0 (75.0, 90.0)	75.0 (62.5, 87.5)
Follow up	70.0 (70.0, 80.0)	80.0 (70.0, 80.0)	70.0 (70.0, 80.0)	75.0 (70.0, 87.5)
% of change	−1.3	−3.5	8.6	−3.3
Weight (kg)				
Basal	86.8 (68.0, 107.0)	75.4 (68.5, 83.1)	85.5 (74.5, 95.7)	73.7 (61.8, 82.9)
Follow up	84.9 (70.2, 107.5)	75.2 (68.5, 80.5)	70.0 (70.0, 80.0)	82.2 (67.1, 84.9)
% of change	0.87	0.40	1.26	−1.68
BMI (kg/m^2^)				
Basal	32.2 (27.9, 39.0)	28.3 (27.9, 35.0)	31.7 (28.8, 37.6)	28.9 (26.3, 34.3)
Follow up	32.0 (27.4, 39.2)	28.3 (28.1, 33.5)	32.0 (28.9, 36.2)	29.9 (27.1, 36.4)
% of change	0.87	0.45	1.27	−1.69
WC (cm)				
Basal	108.8 (89.5, 119.8)	103.0 (95.5, 118.0)	103.5 (98.5, 114.9)	94.4 (89.8, 104.3)
Follow up	102.3 (92.5, 116.5)	98.0 (90.5, 116.0)	104.5 (99.0, 110.0)	94.2 (85.7, 105.5)
% of change	5.50	1.35	−1.15	6.1
Muscular strength				
Basal	23.8 (20.0, 30.9)	27.8 (17.3, 32.5)	23.3 (20.3, 27.7)	19.8 (15.8, 30.0)
Follow up	24.0 (20.6, 34.1)	29.0 (18.0, 36.2)	21.3 (19.0, 28.2)	19.1 (13.7, 31.2)
% of change (*)	−6.16	−3.52	4.51	4.04

Quantitative data are shown as mean ± DE, or median (p25–p75), and qualitative data are shown as *n* (%). Groups: E, exercise isokinetic; D, Mediterranean diet; D + E, Mediterranean diet + exercise isokinetic; and C, control. Bartlet test for unequal variances *p* = 0.02. (*) *p* < 0.05 Basal vs. follow-up estimated by Wilcoxon Test.

**Table 3 nutrients-17-00256-t003:** Changes in biochemical characteristics of the study population.

	D + E (*n* = 12)	E (*n* = 8)	D (*n* = 14)	C (*n* = 8)
Glucose				
Basal	128 (121.5–151.5)	100 (93–153)	110 (93.5–147.5)	125.5 (104–147)
Follow up	128 (120–152)	106 (103–183)	114 (83–153.5)	114.5 (86.7–114.5)
% of change (_95%_CI)	0 (−53.8, 83)	+6 (−19.3, 94.4)	+3.6 (−11.3, 15) *	−8.7 (−84.6, 0)
Triglycerides				
Basal	141.5 (107–176)	115 (101–242.5)	120 (104–130.5)	134.5 (106.5–166.5)
Follow up	139 (102–176)	134 (120–150)	107 (100.5–135.5)	149 (145–206)
% of change (_95%_CI)	−1.7 (−15.1, 9.8)	+16.5 (−19.1, 29.6)	−10.8 (−17.2, 22.0)	+10.7 (−5.7, 13.3)
IL−1b				
Basal	215.8 (186.8–249.4)	200.2 (173.4–237.6)	223.5 (192.1–313.1)	280.7 (220.1–297.7)
Follow up	210.6 (196.8–245.9)	273.9 (251.1–296.6)	204.9 (200.4–276.2)	229.1 (204.5–295.9)
% of change (_95%_CI)	−0.24 (−33.2, 8.5)	+36.8 (−14.8, 91.0)	−8.3 (−18.3, 79.2)	−18.4 (−25.0, 21.3)
Resistin				
Basal	902.3 (603.0–1074.0)	1213.0 (740.4–1939.0)	1057.0 (750.1–1348.0)	820.0 (521.4–1209.0)
Follow up	625.4 (519.8–754.8) *	573.5 (408.5–832.4) *	762.5 (491.0–790.8)	650.9 (516.1–761.4)
% of change (_95%_CI)	−16.7 (−53.5, 28.0)	−32.9 (−123.1, 92.6)	−1.7 (−23.1, 29.1)	−7.6 (−28.0, 38.1)
IL-10				
Basal	431.5 (197.0–477.3)	172.4 (106.5–481.4)	398.7 (120.6–466.5)	283.6 (186.3–513.6)
Follow up	398.4 (148.4–458.7)	239.9 (115.4–416.0)	312.5 (186.0–397.2)	382.6 (197.4–465.5)
% of change (_95%_CI)	−2.37 (−24.6, 26.9) **	+0.97 (−13.5, 37.9)	−11.89 (−31.3, 50.5) **	+9.2 (−37.3, 9.8)
Adiponectin				
Basal	3.4 (2.3–5.4)	4.5 (3.4–5.6)	2.5 (2.1–5.3)	2.4 (1.7–3.4)
Follow up	5.4 (1.6–6.8)	7.2 (5.7–7.7) *	4.8 (1.5–6.5)	2.5 (1.6–5.5)
% of change (_95%_CI)	+39.6 (−18.5, 71.1)	+17.8 (−9.7, 67.1)	+9.7 (−37.8, 45.8)	+3.1 (−74.1, 46.8)

Quantitative data are shown as mean ± DE, or median (p25–p75), and qualitative data are shown as *n* (%). Groups: E, exercise isokinetic; D, Mediterranean diet; D + E, Mediterranean diet + exercise isokinetic; and C, control. (*) *p* < 0.05 Basal vs. follow-up estimated by Wilcoxon Test. (**) *p* < 0.05 Interventions vs. control group estimated with Kruskal–Wallis using the Bonferroni correction. A _95%_CI-95% confidence interval is shown. Cohen´s D effect sizes: Resistin: D + E vs. E (large effect); D + E vs. D (large effect); and D vs. C (large effect). D + E vs. C (medium effect). E vs. D (small effect). IL-10: D vs. C (large effect); D + E vs. D (medium effect); D + E vs. C (medium effect). Adiponectin: D + E vs. C (large effect); E vs. D (large effect); E vs. C (large effect). D + E vs. D (medium effect); D vs. C (medium effect). Small effect > 0.2, medium effect > 0.5, large effect > 0.8.

## Data Availability

The datasets are not publicly available due to privacy policies of the hospital and patient’s sensitive information, but are available from the corresponding author on reasonable request.

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
