# Peer review of "Effect of Mediterranean Diet in Combination with Isokinetic Exercise Therapy on Body Composition and Cytokine Profile in Patients with Metabolic Syndrome"

_nutrients, 2025, doi:10.3390/nu17020256_

Round 1
Reviewer 1 Report
Comments and Suggestions for Authors
Effect of mediterranean diet in combination with isokinetic ex-2 ercise therapy on body composition and cytokine profile in pa-3 tients with metabolic syndrome
All abbreviations used in abstract should be explained (PIC)
The number of patients enrolled in the study is quite small; justification should be provided, since results based on such a group cannot be extrapolated in order to formulate a conclusion
Details should be provided regarding the assessment of biochemical/immunological parameters. Authors state The following biochemical variables 146 were evaluated: circulating levels of cytokines: IL -1, IL-10, sarcolipin, adiponectin by the 147 Thermo Fisher ELISA method. The exact protocols/name of manufacturers/type of methods should be presented for each of the assays
A justification for weight increase of the control group should be privoded
A reference to table 3 should be included in section 3.2 Cytokines. In this section autjors state “In addition, the diet and diet/exercise com- 164 bination groups showed a more significant decrease in IL-10, as compared to control 165 group.” Results show a 9,2% increase in control and a 2,32% decrease in D+E. So, the statement should be corrected since a decrease cannot be compared to an increase in this setting
The Discussion section should be re-written to eliminate repetitions. Authors state “exercise provided an anti-inflamma-214 tory summation effect, mainly given by resistin and adiponectin; whereas diet was related 215 to reduction of IL-10.”and also “we observed that exercise decreased resistin and increased adi-223 ponectin.”. Another example “In general, it can be said that exercise induced a 234 global anti-inflammatory effect, which suggest potential benefits from isokinetic exercise, 235 particularly in population with cardiometabolic risk, due to modification of inflammatory 236 mediators” – this is obvious
Also, authors should remember the very small number of patients included in their study before performing a discussion based on a comparison with other studies from literature.
The conclusion of the study is represented by a phrase that has been repeated few times in the manuscript, showing the results obtained. So, this should be re-written as a conclusion not a repetition of results
Language- should be revised. Some examples below
…and it ranges between 14% and 50% in other 63 populations inculind Latin-American, with higher rates in older age groups and urban 64 populations
Patients diagnosed with MS according to 91 NCEP-ATP III classification [11].
Patients were excluded if chronic renal failure (KDIGO 92 3 or higher), cancer, HIV, chronic heart and/or liver failure and COPD, any limitation to 93 perform physical activity or to already follow a structured exercise plan at the time of 94 recruitment.
After sampling, the sam-144 ples were placed on ice and centrifuged 30 minutes after sampling.
Author Response
REVIEWER 1, COMMENT 1. All abbreviations used in abstract should be explained (PIC).
RESPONSE 1. The authors acknowledge reviewer´s comment.
All abbreviations were revised, explained and corrected.
REVIEWER 1, COMMENT 2. The number of patients enrolled in the study is quite small; justification should be provided, since results based on such a group cannot be extrapolated in order to formulate a conclusion.
RESPONSE 2. The authors acknowledge reviewer´s comment.
We agree that sample size is small, and might not support the magnitude and significance of our results; however, the effect size estimated by Cohen´s D for the cytokines differences were calculated using the software G*Power® by the two-tailed Wilcoxon-Mann-Whitney test and individually comparing the groups:
- IL-10: Group 1 vs Group 2; effect size <0.1, Group 1 vs Group 3; effect size=0.5, Group 1 vs Group 4; effect size=0.7, Group 2 vs Group 3; effect size<0.1, Group 2 vs Group 4; effect size<0.1, Group 3 vs Group 4; effect size >1
- Resistin: Group 1 vs Group 2; effect size>1, Group 1 vs Group 3; effect size=0.9, Group 1 vs Group 4; effect size=0.7, Group 2 vs Group 3; effect size=0.4, Group 2 vs Group 4; effect size<0.1, Group 3 vs Group 4; effect size >1
- Adiponectin: Group 1 vs Group 2; effect size<0.1, Group 1 vs Group 3; effect size=0.7, Group 1 vs Group 4; effect size>1, Group 2 vs Group 3; effect size=0.9, Group 2 vs Group 4; effect size>1, Group 3 vs Group 4; effect size=0.6
In accordance, the following paragraph was modified in methods:
“…Statistical analysis. For continuous variables, the results were presented in mean ± standard deviation, or P50 (P25, P75), depending on normality of distribution according to the Shapiro-Wilk test. Likewise, categorical variables were resumed as n (%). Inferential analyses to evaluate baseline data were performed either by ANOVA or Kruskal-Wallis tests according to the normality of distribution. The Chi-square test was used to compare categorical variables. After the intervention, Wilcoxon test and Kruskal Wallis (with U-Mann-Whitney U test using the Bonferroni correction if applied) were performed to compare intra- and inter-group differences, respectively. Hotelling’s T2 test was used to evaluate the changes in BIVA vectors (resistance and reactance) in each group. The effect size for continuous variables, was estimated by Cohen D test using software G*Power® (11). A p-value of < 0.05 was considered statistically significant…”
And the footnote of Table 3 was modified as follows:
“…(*) p<0.05 Basal vs. follow up estimated by Wilcoxon Test. (**) p<0.05 Interventions vs. control group estimated with Kruskal-Wallis using the Bonferroni correction. Cohen´s D effect size: (†) small effect >0.2, (‡) medium effect >0.5, (††) large effect >0.8. Resistin: †† D+E vs E, D+E vs D, and D vs C; ‡ D+E vs C; † E vs D. IL-10: †† D vs C; ‡ D+E vs D, D+E vs C. Adiponectin: †† D+E vs C, E vs D, E vs C; ‡ D+E vs D, D vs C…”
In addition, the discussion was enriched as follows:
“…Although potential bias related to the small number of patients, interventions induced significant changes in cytokine profiles like resistin, adiponectin and IL-10 as estimated by a medium and large effect sizes, regardless the sample size. Regarding changes in plasma resistin and adiponectin, previous studies have shown conflicting results about the effect of exercise on these adipokines…”
Related reference was added:
- Faul F, Erdfelder E, Lang AG, Buchner A. G*Power 3: a flexible statistical power analysis program for the social, behavioral, and biomedical sciences. Behav Res Methods. 2007;39(2):175-91.
REVIEWER 1, COMMENT 3. Details should be provided regarding the assessment of biochemical/immunological parameters. Authors state: The following biochemical variables were evaluated: circulating levels of cytokines: IL -1, IL-10, sarcolipin, adiponectin by the Thermo Fisher ELISA method. The exact protocols/name of manufacturers/type of methods should be presented for each of the assays.
RESPONSE 3. The authors appreciate reviewer’s comment.
The following paragraph was completed with detailed information: “Circulating levels of cytokines: IL -1, IL-10, adiponectin and sarcolipin, were determined by colorimetric ELISA Assay Kits (IL1b cat.BMS224-2TEN; IL-10 cat.BMS215-2TEN, Adiponectin cat.BMS2032-2 ELISA kits Thermo Fisher, Massachusets, USA; and sarcolipin cat.027956 Sarcolipin (SLN) BioAssay ELISA Kit US Biological Life Sciences Massachusetts, USA); following provider’s instructions. Briefly, plasma sample is added to pre-coating microplate with a capture antibody. Then, a secondary antibody conjugated with enzyme, is added. Upon adding a substrate, the enzyme produces a color change, which was quantified using a spectrophotometer to determine the concentration of the target molecule.”
REVIEWER 1, COMMENT 4. A justification for weight increase of the control group should be provided.
RESPONSE 4. The authors appreciate reviewer’s comment.
The following paragraph was added in the discussion section:
“In addition, increased in body weight observed in the control group may be explained because the general recommendations about diet and exercise given in this group…”
REVIEWER 1, COMMENT 5. A reference to table 3 should be included in section 3.2 Cytokines.
RESPONSE 5. The authors appreciate reviewer’s comment. Reference to Table 3 was added.
REVIEWER 1, COMMENT 6. In this section authors state “In addition, the diet and diet/exercise combination groups showed a more significant decrease in IL-10, as compared to control group.” Results show a 9,2% increase in control and a 2,32% decrease in D+E. So, the statement should be corrected since a decrease cannot be compared to an increase in this setting.
RESPONSE 6. The authors appreciate reviewer’s comment.
We agree. The statement was modified as:
“….In addition, the diet and diet/exercise combination groups showed a significant decrease in IL-10; while an increasing trend was observed in the control group (Table 3).”
REVIEWER 1, COMMENT 7. The Discussion section should be re-written to eliminate repetitions. Authors state “exercise provided an anti-inflammatory summation effect, mainly given by resistin and adiponectin; whereas diet was related to reduction of IL-10.”and also “we observed that exercise decreased resistin and increased adiponectin.”. Another example “In general, it can be said that exercise induced a global anti-inflammatory effect, which suggest potential benefits from isokinetic exercise, particularly in population with cardiometabolic risk, due to modification of inflammatory mediators” – this is obvious.
RESPONSE 7. The authors appreciate reviewer’s comment.
We agree. Some sentences were modified:
“….we observed that exercise decreased resistin and increased adiponectin.” was modified as: “interventions induced significant changes in plasma resistin and adiponectin….”
“In general, it can be said that exercise induced a global anti-inflammatory effect, which suggest potential benefits from isokinetic exercise, particularly in population with cardiometabolic risk, due to modification of inflammatory mediators…” was modified as “…In general, it can be said that exercise induced a global anti-inflammatory effect, which suggest potential benefits particularly in population with cardiometabolic risk…”
REVIEWER 1, COMMENT 8. Also, authors should remember the very small number of patients included in their study before performing a discussion based on a comparison with other studies from literature.
RESPONSE 8. The authors appreciate reviewer’s comment.
We agree. Some sentences were modified:
“Likewise, interventions induced changes in metabolic health, which further related with cytokines profiles. For instance, a decrease in triglycerides and systolic blood pressure associated with concomitant increase in adiponectin and reduction of resistin, respectively; whereas a lack of decrease of glucose was related to a significant reduction of IL-10. Although potential bias related to the small number of patients, interventions induced significant changes in cytokine profiles like resistin, adiponectin and IL-10 as estimated by a medium and large effect sizes, regardless the sample size. Regarding changes in plasma resistin and adiponectin, previous studies have shown conflicting results about the effect of exercise on these adipokines …”
REVIEWER 1, COMMENT 9. The conclusion of the study is represented by a phrase that has been repeated few times in the manuscript, showing the results obtained. So, this should be re-written as a conclusion not a repetition of results.
RESPONSE 9. The authors appreciate reviewer’s comment.
The conclusion was modified as follows: “In conclusion, exercise and Mediterranean diet induced changes in cytokines like re-sistin, adiponectin and IL-10, that suggest an anti-inflammatory summation effect; which may help to explain their benefits in decreasing cardiometabolic risk.”
REVIEWER 1, COMMENT 10. Language- should be revised. Some examples below
…and it ranges between 14% and 50% in other 63 populations inculind Latin-American, with higher rates in older age groups and urban 64 populations
Patients diagnosed with MS according to NCEP-ATP III classification.
Patients were excluded if chronic renal failure (KDIGO 3 or higher), cancer, HIV, chronic heart and/or liver failure and COPD, any limitation to perform physical activity or to already follow a structured exercise plan at the time of recruitment.
After sampling, the samples were placed on ice and centrifuged 30 minutes after sampling.
RESPONSE 10. The authors appreciate reviewer’s comment. Language mistakes and typos were revised and corrected.

Reviewer 2 Report
Comments and Suggestions for Authors
The manuscript addresses an important and timely topic, but several areas could be improved for clarity and scientific rigor. First, the study's objective is clear, but it would benefit from a more explicit hypothesis. The authors should clearly state whether they expect one intervention (Mediterranean diet or iso-kinetic exercise) to be more effective than the other or if they are primarily interested in the combined effect. This would help guide the interpretation of the results. Additionally, while choosing biomarkers such as resistin, adiponectin, and IL-10 is relevant, the authors should provide a more explicit rationale for selecting these markers over others. Explaining why these biomarkers were chosen would add depth to the manuscript.The study duration of 12 weeks and the relatively small sample size (42 patients) may limit the generalizability of the findings. A more extended follow-up period would allow for assessing long-term effects, and a larger sample size could improve statistical power.
Furthermore, while the study focuses on biomarkers and body composition, additional clinical outcomes such as insulin sensitivity, blood pressure, and lipid profiles would provide a more comprehensive assessment of the interventions' effects on metabolic syndrome. The statistical analysis section would also benefit from more detailed explanations. The authors should clarify whether multiple comparisons correction was applied or if confounding factors were controlled for. Including confidence intervals and effect sizes would help readers better understand the magnitude and significance of the results.
Some results, such as the reduction in resistin and the decrease in IL-10, are interesting, but the manuscript would benefit from a more profound interpretation. How do cytokine changes relate to improvements in metabolic health or cardiovascular risk? The changes in body composition should also be discussed regarding their clinical relevance. Do the observed changes in body composition signify meaningful health improvements for the participants? The practical implications of the findings could be better highlighted. For instance, how feasible are these interventions in real-world settings? Could they be applied to a broader population? The manuscript could also be more transparent in its language and presentation. For example, the term "resistance values" should be explained to readers who may not be familiar with the measurement techniques. The discussion section could be more.
Author Response
REVIEWER 2, COMMENT 1. The manuscript addresses an important and timely topic, but several areas could be improved for clarity and scientific rigor. First, the study's objective is clear, but it would benefit from a more explicit hypothesis.
RESPONSE 1. The authors appreciate reviewer’s comment.
An hypothesis was added at the introduction section: “We hypothesized that each intervention is able to induce different effects on BC and/or PIC, which result in a joined benefit for patients with cardiometabolic risk.”
REVIEWER 2, COMMENT 2. The authors should clearly state whether they expect one intervention (Mediterranean diet or iso-kinetic exercise) to be more effective than the other or if they are primarily interested in the combined effect. This would help guide the interpretation of the results.
RESPONSE 2. The authors appreciate reviewer’s comment.
As stated in the hypothesis, interventions were expected to exert different effects on body composition and to differentially affect pro-inflammatory mediators, while non-particularly superiority was claimed. Then, we think that the hypothesis added reflects this thought.
REVIEWER 2, COMMENT 3. Additionally, while choosing biomarkers such as resistin, adiponectin, and IL-10 is relevant, the authors should provide a more explicit rationale for selecting these markers over others. Explaining why these biomarkers were chosen would add depth to the manuscript.
RESPONSE 3. The authors appreciate reviewer’s comment.
The following paragraph was added to discussion: “Furthermore, cytokines like IL-1, IL-10, resistin and adiponectin are involved in adipose tissue metabolism, BC and regulation of low degree inflammation; therefore, representing potential prognostic/intervention targets between population with cardiometabolic risk”.
REVIEWER 2, COMMENT 4. The study duration of 12 weeks and the relatively small sample size (42 patients) may limit the generalizability of the findings. A more extended follow-up period would allow for assessing long-term effects, and a larger sample size could improve statistical power.
RESPONSE 4. The authors acknowledge reviewer´s comment.
We agree that sample size is small, and might not support the magnitude and significance of our results; however, the effect size estimated by Cohen´s D for the cytokines differences were calculated using the software G*Power® by the two-tailed Wilcoxon-Mann-Whitney test and individual comparison of groups:
- IL-10: Group 1 vs Group 2; effect size <0.1, Group 1 vs Group 3; effect size=0.5, Group 1 vs Group 4; effect size=0.7, Group 2 vs Group 3; effect size<0.1, Group 2 vs Group 4; effect size<0.1, Group 3 vs Group 4; effect size >1
- Resistin: Group 1 vs Group 2; effect size>1, Group 1 vs Group 3; effect size=0.9, Group 1 vs Group 4; effect size=0.7, Group 2 vs Group 3; effect size=0.4, Group 2 vs Group 4; effect size<0.1, Group 3 vs Group 4; effect size >1
- Adiponectin: Group 1 vs Group 2; effect size<0.1, Group 1 vs Group 3; effect size=0.7, Group 1 vs Group 4; effect size>1, Group 2 vs Group 3; effect size=0.9, Group 2 vs Group 4; effect size>1, Group 3 vs Group 4; effect size=0.6
In accordance, the following paragraph was modified in methods:
“…Statistical analysis. For continuous variables, the results were presented in mean ± standard deviation, or P50 (P25, P75), depending on normality of distribution according to the Shapiro-Wilk test. Likewise, categorical variables were resumed as n (%). Inferential analyses to evaluate baseline data were performed either by ANOVA or Kruskal-Wallis tests according to the normality of distribution. The Chi-square test was used to compare categorical variables. After the intervention, Wilcoxon test and Kruskal Wallis (with U-Mann-Whitney U test using the Bonferroni correction if applied) were performed to compare intra- and inter-group differences, respectively. Hotelling’s T2 test was used to evaluate the changes in BIVA vectors (resistance and reactance) in each group. The effect size for continuous variables, was estimated by Cohen D test using software G*Power® (11). A p-value of < 0.05 was considered statistically significant…”
And the footnote of Table 3 was modified as follows:
“…(*) p<0.05 Basal vs. follow up estimated by Wilcoxon Test. (**) p<0.05 Interventions vs. control group estimated with Kruskal-Wallis using the Bonferroni correction. Cohen´s D effect size: (†) small effect >0.2, (‡) medium effect >0.5, (††) large effect >0.8. Resistin: †† D+E vs E, D+E vs D, and D vs C; ‡ D+E vs C; † E vs D. IL-10: †† D vs C; ‡ D+E vs D, D+E vs C. Adiponectin: †† D+E vs C, E vs D, E vs C; ‡ D+E vs D, D vs C…”
In addition, the discussion was enriched as follows:
“…Although potential bias related to the small number of patients, interventions induced significant changes in cytokine profiles like resistin, adiponectin and IL-10 as estimated by a medium and large effect sizes, regardless the sample size. Regarding changes in plasma resistin and adiponectin, previous studies have shown conflicting results about the effect of exercise on these adipokines…”
Related reference was added:
- Faul F, Erdfelder E, Lang AG, Buchner A. G*Power 3: a flexible statistical power analysis program for the social, behavioral, and biomedical sciences. Behav Res Methods. 2007;39(2):175-91.
REVIEWER 2, COMMENT 5. Furthermore, while the study focuses on biomarkers and body composition, additional clinical outcomes such as insulin sensitivity, blood pressure, and lipid profiles would provide a more comprehensive assessment of the interventions' effects on metabolic syndrome.
RESPONSE 5. The authors appreciate reviewer’s comments.
Metabolic health was denoted by changes in triglycerids, glucose and arterial pressure,. And were related to changes in cytokines. The following paragraphs were added to discussion:
“Interventions induced changes in metabolic health, which further related with cytokines profiles. For instance, a decrease in triglycerides and systolic blood pressure was associated with concomitant increase in adiponectin and reduction of resistin, respectively; whereas a lack of decrease of glucose was related to a significant reduction of IL-10…”
REVIEWER 2, COMMENT 6. The statistical analysis section would also benefit from more detailed explanations. The authors should clarify whether multiple comparisons correction was applied or if confounding factors were controlled for. Including confidence intervals and effect sizes would help readers better understand the magnitude and significance of the results.
RESPONSE 6. The authors acknowledge reviewer´s comment.
The following paragraphs was modified in methods:
“…Statistical analysis. For continuous variables, the results were presented in mean ± standard deviation, or P50 (P25, P75), depending on normality of distribution according to the Shapiro-Wilk test. Likewise, categorical variables were resumed as n (%). Inferential analyses to evaluate baseline data were performed either by ANOVA or Kruskal-Wallis tests according to the normality of distribution. The Chi-square test was used to compare categorical variables. After the intervention, Wilcoxon test and Kruskal Wallis (with U-Mann-Whitney U test using the Bonferroni correction if applied) were performed to compare intra- and inter-group differences, respectively. Hotelling’s T2 test was used to evaluate the changes in BIVA vectors (resistance and reactance) in each group. The effect size for continuous variables, was estimated by Cohen D test using software G*Power® (11). A p-value of < 0.05 was considered statistically significant…”
In order to better support the magnitude and significance of our results the effect size was estimated by Cohen´s D for the cytokines differences, using the software G*Power® by the two-tailed Wilcoxon-Mann-Whitney test and individual comparison of groups:
- IL-10: Group 1 vs Group 2; effect size <0.1, Group 1 vs Group 3; effect size=0.5, Group 1 vs Group 4; effect size=0.7, Group 2 vs Group 3; effect size<0.1, Group 2 vs Group 4; effect size<0.1, Group 3 vs Group 4; effect size >1
- Resistin: Group 1 vs Group 2; effect size>1, Group 1 vs Group 3; effect size=0.9, Group 1 vs Group 4; effect size=0.7, Group 2 vs Group 3; effect size=0.4, Group 2 vs Group 4; effect size<0.1, Group 3 vs Group 4; effect size >1
- Adiponectin: Group 1 vs Group 2; effect size<0.1, Group 1 vs Group 3; effect size=0.7, Group 1 vs Group 4; effect size>1, Group 2 vs Group 3; effect size=0.9, Group 2 vs Group 4; effect size>1, Group 3 vs Group 4; effect size=0.6
And the footnote of Table 3 was modified as follows:
“…(*) p<0.05 Basal vs. follow up estimated by Wilcoxon Test. (**) p<0.05 Interventions vs. control group estimated with Kruskal-Wallis using the Bonferroni correction. Cohen´s D effect size: (†) small effect >0.2, (‡) medium effect >0.5, (††) large effect >0.8. Resistin: †† D+E vs E, D+E vs D, and D vs C; ‡ D+E vs C; † E vs D. IL-10: †† D vs C; ‡ D+E vs D, D+E vs C. Adiponectin: †† D+E vs C, E vs D, E vs C; ‡ D+E vs D, D vs C…”
In addition, the discussion was enriched as follows:
“…Although potential bias related to the small number of patients, interventions induced significant changes in cytokine profiles like resistin, adiponectin and IL-10 as estimated by a medium and large effect sizes, regardless the sample size. Regarding changes in plasma resistin and adiponectin, previous studies have shown conflicting results about the effect of exercise on these adipokines…”
REVIEWER 2, COMMENT 7. Some results, such as the reduction in resistin and the decrease in IL-10, are interesting, but the manuscript would benefit from a more profound interpretation. How do cytokine changes relate to improvements in metabolic health or cardiovascular risk? The changes in body composition should also be discussed regarding their clinical relevance. Do the observed changes in body composition signify meaningful health improvements for the participants? The practical implications of the findings could be better highlighted. For instance, how feasible are these interventions in real-world settings? Could they be applied to a broader population? The manuscript could also be more transparent in its language and presentation. For example, the term "resistance values" should be explained to readers who may not be familiar with the measurement techniques. The discussion section could be more.
RESPONSE 7. The authors appreciate reviewer’s comments.
The following paragraphs were added to discussion:
“Interventions induced changes in metabolic health, which further related with cytokines profiles. For instance, a decrease in triglycerides and systolic blood pressure was associated with concomitant increase in adiponectin and reduction of resistin, respectively; whereas a lack of decrease of glucose was related to a significant reduction of IL-10…”
“Regarding the effect of interventions on BC, the resistance, a measure denoting the opposition to current given by the amount of cells and water, increased in all groups that performed exercise, suggesting an effect on muscle mass. Interestingly, a lower increase in resistance was observed in the group with MedDiet, whereas reactance, other measure of BC given by the opposition to current caused by energy accumulation in cell membranes, was significantly increased in the control group. Possible explanations include a higher compliance with the exercise protocol. Consistently, Tinsley et al. demonstrated changes in resistance and reactance values in healthy women who performed resistance training (17).”
“..In the other hand, we consider that exercise-induced changes obtained in the present study are feasible to achieve in a real-world scenario; since OMS-recommended exercise routine, which was performed by the control group, rendered similar results as the isokynetic exercise protocol, in terms of metabolic health, BC and cytokines profile. However, MedDiet was a bit more difficult to achieve, mainly due to local cultural feeding habits; whereas a tropicalized MedDiet may be a reasonable option.…”
Related reference was added:
- Tinsley, G. M., Harty, P. S., Moore, M. L., Grgic, J., Silva, A. M., & Sardinha, L. B. Changes in total and segmental bioelectrical resistance are correlated with whole-body and segmental changes in lean soft tissue following a resistance training intervention. Journal of the International Society of Sports Nutrition. 2019. 16(1):58.

Round 2
Reviewer 1 Report
Comments and Suggestions for Authors
Authors responded to comments and improved the manuscript
Reviewer 2 Report
Comments and Suggestions for Authors
The authors have made significant treads in addressing the reviewers' feedback, allowing the manuscript to be acceptable for publication, after a few minor improvements. First, while the practical implications are discussed, elaborating a bit on the clinical significance of body composition changes and cytokine profiles in relation to long-term health outcomes could consistently enhance the manuscript quality and importance in the field. Additionally, including confidence intervals for the major findings would provide stronger context and improve the reliability of the study. Finally, the discussion could be strengthened by clearly linking cytokine changes, metabolic health, and cardiovascular risk, along with a brief comparison to current studies.
Comments on the Quality of English LanguageThe language in the manuscript is generally understandable, but refining some sections to improve clarity and readability will enhance its quality. These revisions are minor and should be addressed as part of the overall revision process.
